# General ADP-Ribosylation Mechanism Based on the Structure of ADP-Ribosyltransferase–Substrate Complexes

**DOI:** 10.3390/toxins16070313

**Published:** 2024-07-11

**Authors:** Hideaki Tsuge, Noriyuki Habuka, Toru Yoshida

**Affiliations:** 1Faculty of Life Sciences, Kyoto Sangyo University, Kyoto 6038555, Japan; 2Faculty of Sciences, Japan Women’s University, Tokyo 1120015, Japan

**Keywords:** ADP-ribosylation, ADP-ribosyltransferase, complex, substrate recognition

## Abstract

ADP-ribosylation is a ubiquitous modification of proteins and other targets, such as nucleic acids, that regulates various cellular functions in all kingdoms of life. Furthermore, these ADP-ribosyltransferases (ARTs) modify a variety of substrates and atoms. It has been almost 60 years since ADP-ribosylation was discovered. Various ART structures have been revealed with cofactors (NAD^+^ or NAD^+^ analog). However, we still do not know the molecular mechanisms of ART. It needs to be better understood how ART specifies the target amino acids or bases. For this purpose, more information is needed about the tripartite complex structures of ART, the cofactors, and the substrates. The tripartite complex is essential to understand the mechanism of ADP-ribosyltransferase. This review updates the general ADP-ribosylation mechanism based on ART tripartite complex structures.

## 1. Introduction

Pathogenic bacteria produce various protein toxins and effectors that covalently modify host proteins to affect their functions or mimic host protein functions to interfere with host–cell regulatory processes [1]. ADP-ribosylation is a ubiquitous modification of not only proteins but also other targets, such as nucleic acids, that regulates various cellular functions in all kingdoms of life [2]. ADP-ribosyltransferases (ARTs) transfer an ADP-ribose moiety from nicotinamide adenine dinucleotide (NAD^+^) to a target biomolecule to generate an ADP-ribosylated target and nicotinamide. The bulky and negatively charged ADP-ribose moiety alters the target’s function by sterically blocking interactions with partner molecules, inducing conformational changes, or creating docking sites for new interactions [3]. In addition to their function as bacterial toxins, ARTs have been implicated in a wide range of processes in various organisms, including the bacterial toxin–antitoxin system and eukaryotic DNA damage repair, transcription, cell cycle progression, and cell division [4]. 

ARTs are subdivided into two classes based on their conservation of three significant motifs. One class is the His-Tyr-Glu class (ARTD), which is related to the first identified ART, the diphtheria toxin (DT) [2]. Honjo et al. showed that NAD is required for the toxin’s action due to ART reaction-modifying elongation factor 2 [5,6]. Specifically, both DT and *Pseudomonas aeruginosa* exotoxin ADP-ribosylate diphthamide require NAD to activate a modified histidine in elongation factor 2 [5,7]. The other class is the Arg-Ser-Glu class (ARTC), which is related to clostridial toxins, encompassing the following toxins: cholera toxin (CT), which ADP-ribosylates the arginine residues in G proteins; pertussis toxin, which ADP-ribosylates a cysteine residue [8]; *Clostridium botulinum* C3 exoenzyme, which ADP-ribosylates Asn41 on RhoA [9,10]; and *Clostridium perfringens* iota toxin A subunit (Ia), a binary toxin, which ADP-ribosylates Arg177 on actin [11,12]. In humans, there are two ART families that are homologs of these bacterial ARTs: poly(ADP-ribose) polymerase (PARP) [13] and ectoARTs [14,15], which belong to ARTD and ARTC, respectively. 

Several lines of evidence have shown that the ART reaction proceeds through an SN1-like transition state via an oxocarbenium ion (Figure 1A). In the case of DT, the transition state model most consistent with the isotope kinetic effects is characterized by the loss of bond order in the nicotinamide leaving group, suggesting a dissociative SN1-type reaction [16]. The oxocarbenium ion-like transition state is stabilized by Glu148, which is a key catalytic glutamate of DT [17]. The reaction via oxocarbenium ion-like transition state was also reported in the study of pertussis toxin [18].

It is interesting to note that ARTs in both classes target a variety of amino acids, nucleobases [19,20,21], and antibiotics [22,23]. Furthermore, ARTs modify many different atoms, such as oxygen, nitrogen, and sulfur (Figure 1B; this figure summarizes only the amino acids and bases of DNA targets). These atoms are classified as soft, intermediate, or hard based on the hard and soft acids and bases theory [24], and ARTs can modify them all despite their varying levels of reactivity. How do ARTs select a wide variety of targets while also showing specificity toward many different atoms? This is the general question investigated by ART researchers studying both prokaryotes and eukaryotes, striving to better understand how ARTCs and ARTDs recognize substrates and specify atoms. To answer these questions, structural studies are essential, especially those regarding ART–substrate complexes. The purpose of this review was to compile relevant research related to ART–substrate complexes to gain insights into ADP-ribosylation substrate recognition mechanisms (Table 1).

## 2. Substrate Recognition Mechanism of ARTC: Iota-like Toxin and C3-like Exoenzyme

ARTC structural studies are the primary source of information about the substrate recognition mechanism of ARTs. In particular, comparative studies of iota-like toxin and C3-like exoenzyme have provided essential information about their substrates. Binary toxins are composed of subunit A, actin ART, and subunit B, the protein translocation pore. The subunits of *Clostridium botulinum* C2 toxin [11], *C. perfringens* iota toxin [35], *C. spiroforme* toxin [36], and *C. difficile* toxin [37] modify an arginine residue on actin, subsequently inhibiting actin polymerization. *Bacillus cereus* vegetative insecticidal protein (VIP2) also belongs to this family [38], and these toxins all ADP-ribosylate Arg177 on actin. 

On the other hand, the *Clostridium botulinum* C3 exoenzyme selectively modifies low-molecular-weight GTP-binding proteins RhoA, RhoB, and RhoC [9,39]. C3-like toxins exhibit 35–70% amino acid identity and modify Rho proteins at an asparagine residue (Asn41 on RhoA) [40]. 

Han et al. reported the structures of VIP2 and, subsequently, C3 exoenzyme [38,41]. Though the substrate complex structures were not available for investigation, the authors proposed a hypothesis of substrate recognition through an ADP-ribosylating turn–turn loop (ARTT-loop) as follows: The ARTT-loop of ARTC contains eight amino acids represented as Xxx-Xxx-φ(Hydrophobic residue)-Xxx-Xxx-Glu/Gln-Xxx-Glu (Table 2), consisting of two turns. The first turn contains a hydrophobic residue to recognize the substrate, whereas the second turn contains solvent-exposed Glu/Gln, which is conserved among Rho- and actin-ADP-ribosylating toxins. These structures suggest that Gln on an ARTT-loop may recognize the Asn41 on Rho proteins by forming hydrogen bonds. Similarly, Glu426 on VIP2, which is in the same position, may recognize the substrate by forming salt bridges with the Arg177 guanidinium group of actins, rather than facilitating a nucleophilic attack by the guanidinium, as initially proposed (Han et al. (1999) [38]). In the early 2000s, many structures of C3-like exoenzymes and iota-like toxins were characterized and reported [12,42,43,44,45]. These studies revealed that the two molecules have the same basic structural framework and similar NAD conformations. However, these studies did not show how these enzymes recognize the substrate. 

Our research group investigated this knowledge gap and revealed the structures of complexes with full protein substrates (C3–RhoA and Ia-actin) (Figure 2 and Figure 3) [25,26]. To understand the mechanism using *Bacillus cereus* C3 exoenzyme, three RhoA complex structures were reported: C3–RhoA (GTP), C3–RhoA (GTP)–NADH, and C3–RhoA (GDP)–NADH [26]. C3 recognizes the switch I, switch II, and inter-switch regions in RhoA. Interestingly, it was shown that C3 binding induces the same conformational change in the switch I and II regions in NADH–C3–RhoA (GTP) and NADH–C3–RhoA (GDP) complexes, respectively, which is why C3 ADP-ribosylates both forms of RhoA. This plasticity enables the Asn41 residue to move closer to the ARTT-loop of C3, which is a critical interaction that takes place between RhoA and C3. The structure shows the three-way relationship between Gln183 in the GlnXxxGlu motif (C3), a modified Asn41 residue (RhoA), and NC1 of NAD(H). The distance between ND2 of Asn41 and NC1 of NAD(H) is 3.6 Å, suggesting a possible direct attack by the nitrogen atom of Asn41 (nucleophile) (Figure 2). 

Another important question is the following: how is Arg177 on actin recognized by the ARTT-loop of the iota-like toxin? Within the NAD^+^–Ia-actin complex, the distance between the nucleophile (Arg177) and the electrophile (NC1 on N-ribose) is 8.4 Å [25] (Figure 3). Therefore, it seems impossible for a direct attack to occur. More recent studies using Ia-actin crystals revealed the pre- and post-reaction complex structures through NAD soaking experiments [46]. Based on this structural information, the strain-alleviation model for ADP-ribosylation via the oxocarbenium cation was proposed as follows (Figure 4): (i) The NAD conformations are shared for all of the ARTs. This specific strained conformation of NAD causes the cleavage of nicotinamide, leading to an SN1 reaction. (ii) When the AMP moiety in the ADP-ribose is gripped by Ia, the phosphoribose moiety in ADP-ribose moves close to the target Arg177 via bond rotation of the pyrophosphate moiety, leading to alleviation of the ADP-ribose conformation. (iii) The nitrogen atom of Arg177 nucleophilically attacks the NC1 carbon atom of ADP-ribose and forms a new bond. Though the strain-alleviation model is widely accepted [47], there is no evidence of a close relationship between Arg177 (actin) and Glu378 in the GluXxxGlu motif (Ia). We re-modeled the torsion angle of the side chains of Arg177 (actin) and Glu378 in the EXE motif to maintain close proximity to each other (our unpublished data), resulting in a distance of 6 Å between Arg177 (actin) and Glu378, which is not close enough to attack the electrophile (NC1 of N-ribose) directly. This critical recognition probably also occurs during the strain-alleviation reaction. 

These complexes between the substrate protein and the ADP-ribosylation enzyme show the importance of ADP-ribosylation specificity. First, enzyme (ART)–substrate interactions are formed. Second, substrate amino acid recognition by the ARTT-loop specifies the reaction. These two general facts were elucidated from the first studies regarding complex structures and mutations of the C3–RhoA complex [26].

## 3. Substrate Recognition Mechanism of Bacterial ARTD

The next important question to answer is the following: how are substrate and atom selectivity achieved in ARTD compared with ARTC? The first crystal structure of ARTD is DT [48], and later studies characterized Glu148 as a key residue in close proximity to the nicotinamide ring of NAD^+^, suggesting that the reaction proceeds via the SN1 mechanism through a positively charged oxocarbenium ion [49]. In 2005, the first structural characterization of an ARTD–substrate complex was reported regarding ExoA-eEF2-NAD^+^ [30,50]. In these structures, it was shown that Asp461 on the flexible L1-loop of ExoA interacts with diphthamide, in addition to the key glutamate Glu553. The authors of this study suggested that one additional step is necessary because the diphthamide N3 nucleophilic atom is still about 10 Å in distance from the NC1 of NAD. Thus, they proposed a transition state model where ExoA undergoes a 3_10_ helix unfolding-like movement upon the diphthamide binding. This pioneering work was the first to reveal pre- and post-reaction states using complex crystals. However, it is still necessary to examine other ARTD complexes in the future. 

ARTDs and ARTCs both have an ARTT-loop, but the ARTT-loops of ARTDs vary widely in length, ranging from short to long, such as ExoA with 7aa, DarT with 44aa, and PARP1 with 37aa [31]. In addition to the variety of ARTT-loop lengths, ARTDs do not have a specific sequence feature, such as the Xxx-Xxx-φ-Xxx-Xxx-Glu/Gln-Xxx-Glu motif found in ARTCs that provide specificity. Furthermore, another amino acid is necessary for ARTDs to show substrate selectivity, such as ExoA Asp461. Thus, it is more difficult to summarize the substrate specificity of ARTDs in general. Nevertheless, to examine this in more detail, we discuss DarT and PARP in later sections of this paper.

## 4. ADP-Ribosylation of Ubiquitin

The post-translational attachment of ubiquitin is one of the most important modifications in eukaryotes. Ubiquitination covalently links C-terminal ubiquitin to a lysine ε-amino on a protein substrate using three enzymes: E1, E2, and E3 [51]. However, bacterial pathogens have evolved elaborate mechanisms to hijack host ubiquitin signaling pathways and evade the immune response. For this purpose, numerous E3 ubiquitin ligases and deubiquitinating enzymes (DUBs) are encoded by bacterial and viral human pathogens. 

The Gram-negative pathogen *Legionella pneumophilla* translocates at least 330 effectors into the host cytosol for survival and replication. Recent studies have revealed that the SidE effector family, represented by SdeA, targets Rab GTPases through a two-step noncanonical ubiquitination pathway involving ADP-ribosylation. First, ubiquitin is mono-ADP-ribosylated at Arg42 by the SdeA mono-ART domain, and then the mono-ADP-ribosylated ubiquitin intermediates (ADPR–ubiquitin) are processed by the phosphodiesterase (PDE) domain [52,53]. Finally, the serine residue of the host target is ubiquitinated via phosphoribosyl (PD)-linked ubiquitin. 

Two research groups have characterized and reported the structures of mART (SdeA and SidE) and ubiquitin complexes, providing information regarding the ADP-ribosylation specificity of ubiquitin arginine residues [28,29] (Figure 5). These mARTs are ARTC class enzymes with the Arg-Ser-Glu feature. The ARTT-loop is different from iota and C3 enzymes, as shown in Table 2. Specifically, it lacks a hydrophobic residue (φ) in turn 1, which normally interacts with a substrate in iota and C3. However, because ubiquitin is a small protein, there is no need to interact using these hydrophobic residues. Both structures show how SdeA and SidE recognize ubiquitin using a very narrow surface. Interestingly, in both structures, the distance between the amino group of Arg72 and the NC1 atom of NAD is 5~6 Å, but previous research has identified that only Arg42, and not Arg72, is the active residue to be modified [52]. Of note, mutating Arg72 also completely abolished ubiquitin modification by SdeA. Accordingly, Arg42 and Arg72 mutants failed to be substrates for SdeA auto-ubiquitination. The author speculates that Arg72 plays a more important role in catalysis by SdeA [52]. Notably, it is insightful to propose that ubiquitin Arg72 may first act as a probe that interacts with the mART domain before movements occur in the side chains of Arg72 and Arg42 during the ADP-ribosylation of ubiquitin [28]. It is unclear why there is no ADP-ribosylation in Arg72, given the proximity between Arg72 and the mART domain. One possible explanation is that the direction of nucleophilic attack against the NC1 domain of NAD may be crucial for ADP-ribosylation. 

Furthermore, it is known that *Chromobacterium violaceum* CteC blocks host ubiquitination through the mono-ADP-ribosylation of ubiquitin at residue Thr66 [54]. In later studies, the structures of ubiquitin-bound complexes were revealed [33,34]. CteC has a ubiquitin-targeting domain in addition to its enzymatic domain, which is key for its interaction with ubiquitin. However, the complex structure revealed that the distance between NC1(NAD) and OG1(Thr66) is too long to allow for an ADP-ribosylation reaction. CteC harbors a unique Asp(134)–Glu(220) catalytic motif but lacks the classic Glu/Gln-Xxx-Glu motif. It was predicted that Asp134 spatially complements the role of the first glutamate/glutamine residue in the classic Glu/Gln-Xxx-Glu motif, and that Asp134 may transiently capture substrate Thr. The author concluded that CteC has a unique Asp-Glu motif and a PARP-like fold with chimeric features from ARTC and ARTD [33].

## 5. ADP-Ribosylation of DNA by ScARP in the Pierisin Family (ARTC)

For a long time, ADP-ribosylation has been considered a post-translational protein modification. However, emerging evidence suggests that DNA ADP-ribosylation is also common. The first DNA-targeting ART was found in pierid butterflies and was thus named pierisin [55,56]. Pierisin ADP-ribosylates calf thymus DNA containing dG-dC, and it also modifies the N2 amino group of the guanine residue in DNA to yield N2-(ADP-ribos-1-yl)-2′-deoxyguanosine [19,57]. Interestingly, the DNA-modifying toxin exists only in some pierid butterflies among a variety of examined Lepidopteran insects. On the other hand, SCO5461 protein (ScARP) from *Streptomyces coelicolor* was identified as an ART that mainly targets mononucleotides and nucleosides and shares 30% homology with pierisin. ScARP ADP-ribosylates deoxyguanosine (dGuo), GMP, dGMP, and cyclic GMP rather than dsDNA, whereas pierisin-1 shows weak ADP-ribosylation activity on dGuo. To date, six pierisins (pierisin-1 to pierisin-6) [57,58,59,60,61,62], ScARP [63], and Scabin [64,65] are considered to belong to the pierisin family, which is classified under the ARTC class. 

Furthermore, CARP-1, which is present in certain kinds of edible clams, also ADP-ribosylates calf thymus DNA to produce N2-(ADP-ribos-1-yl)-2′-deoxyguanosine [20,66]. However, CARP-1 and pierisins share little sequence homology, suggesting that they are not derived from a common ancestral gene [20]. The structure of pierisin has been reported in NAD^+^ and auto-inhibitory forms [67]. Furthermore, the structure of scabin with inhibitors has also been reported [64]. However, the substrate recognition mechanism of these enzymes is still an open question because of the lack of their substrate complexes. 

In 2018, Yoshida and Tsuge reported the structures of apo-ScARP and ScARP-GDP-NADH [27]. These characterizations were the first to elucidate a mechanism by which ARTs identify guanosine (Figure 6). Specifically, the N2 (NH2) and N3 atoms of guanine form hydrogen bonds with OE1 and NE2 of Gln162, respectively. The N2 atom (NH2) of guanine is the acceptor of the ADP-ribosyl moiety, with a distance of 4.0 Å from the NC1 position of N-ribose. In other words, these binding features guarantee guanine specificity and prevent the binding of other bases such as adenine. Gln162 was conserved in pierisin and scabin, so this specificity mechanism was kept in this family. Of note, the relative positions of NAD(H), the acceptor of ADP-ribose, and Gln of the ARTT-loop are precisely the same in two different complex structures of C3 and ScARP. This study showed that these relative position similarities with the ARTT-loop are conserved between protein-targeting ARTs and DNA-targeting ARTs, suggesting their common importance for ADP-ribosylation. 

## 6. ADP-Ribosylation of DNA by DarT2 and DarT1 (ARTD)

Another DNA ART, DarT, was discovered as a bacterial ART toxin encoded in the DarT–DarG toxin–antitoxin system [21]. DarTG systems are widespread among prokaryotes, including many human pathogens. DarTG systems are often encoded next to phage defense elements, suggesting that DarTG systems play a role in providing defense against bacteriophages [68]. DarT (DarT2) has been shown to transfer ADP-ribose from NAD to thymidine bases in single-stranded DNA (ssDNA), specifically at the four-base motif TNTC, and it lacks activity on RNA or protein targets [21]. DarT belongs to the ARTD family, the members of which are similar to PARP. For example, DarT has a long ARTT-loop (44aa), as does PARP (PARP2 has 40aa). 

Schuller et al. revealed the structure of DarT2 (Glu160Ala mutant, a key residue in ADP-ribosylation) using *Thermus* sp. 2.9. High-resolution structural characterizations of NAD^+^ and ART-DNA elucidated the mechanism by which thymidine bases are recognized (nitrogen) (Figure 7A) [31]. Thymidine bases are recognized by His119 and Arg51. This is different from the previously described ARTC base recognition. Furthermore, the author proposed multiple important roles of Arg51, including proton abstraction from N3 of the thymidine base by R51, which ultimately enables ADP-ribose linkage through nucleophilic attack by an oxocarbenium ion in an SN1-type reaction [31]. 

Recently, Schuler et al. biochemically and structurally characterized another DNA ART DarT1, which is the ART of guanosine bases [32] (Figure 7B). They determined the structure of DarT1(Glu152Ala), the mutant of the key ADP-ribosylation residue, in its pre- and post-reaction states. The co-crystallization of DarT1(E152A) with NAD^+^ and ssDNA 5-mer provided insight into the substrate recognition mechanism. Interestingly, the guanine was recognized by Asn104, and its N2 atom is in close proximity with NC1 of NAD^+^. Notably, this is the first study to report that ARTD also ADP-ribosylates guanine in a manner similar to ARTC enzymes, such as ScARP and pierisin. 

Though DarT1 and ScARP differ in their overall structural makeup, the guanine recognition mechanism is the same in both DarT1 and ScARP. In summary, both share the same NAD binding conformation, ART key glutamate (DarT1 model), and guanine specificity mechanism. Even though there are significant differences between short and long ARTT-loops, the guanine specificity of both enzymes is achieved through the positioning of Asn and Gln via N2 and N3 coordination in DarT1 and ScARP, respectively. Of note, we consider the strain-alleviation model applicable based on the structures of the pre- and post-ADP-ribosylation states of DarT, suggesting that the strain-alleviation model is generalizable to both ARTC and ARTD.

## 7. ADP-Ribosylation by Poly(ADP-Ribose) Polymerase (PARP)

The story of the discovery of poly(ADP-ribose) in the mid-1960s has been summarized in references [69,70]. PARPs are alternatively referred to as DT-like ARTs (ARTDs) [71]. In humans, seventeen members of PARPs are known, but most PARPs are mono-ARTs. Several PARPs, including PARP1, PARP2, and tankyrase, are poly(ADP-ribose) polymerases. It was proposed that the triad His-Tyr-Glu is generally an indicator of poly-ADP-ribosylation, and the triad His-Tyr-[Leu/Ile/Val] is a mono-ADP-ribosylation indicator [72]. Notably, PAR levels are tightly regulated in the cell by poly(ADP-ribose) glycohydrolases [73,74]. Each PARP has different domain configurations. For example, PARP1 consists of three zinc finger domains, a BRCA-1 C-terminal domain, a WGR domain, a helical domain (HD), and an ART domain (CAT). These complex domain configurations are involved in PARP activation, which requires an open active site. Unlike bacterial toxins, human PARP1 and PARP2 are activated through binding with damaged DNA. These PARPs use NAD^+^ to modify numerous proteins via mono-ADP-ribosylation and poly-ADP-ribosylation, which are important processes for the subsequent decompaction of chromatin and the recruitment of repair factors [75,76]. 

Several structures of PARPs have been revealed, but structures of the PARP complex with NAD^+^ (NADH) have not been available for a long time. Why are there no crystal structures available to characterize PARPs in a complex with NAD^+^? This question is answered as follows (structural biology of ADP-ribosyltransferase-FAQ [77]): ARTs use NAD^+^ as a co-substrate. They are designed to retain nicotinamide and bind the ADP-ribosyl moiety only loosely, so that it can be passed on to the substrate. This is achieved using a snug nicotinamide binding pocket and an ADP-ribosyl binding site that likely forms only in the enzyme–substrate complex. 

Usually, the catalytic domain of PARP consists of an autoinhibitory helical domain (HD) covering the ART domain, such that HD blocks access to NAD^+^ and the substrate. For a long time, there have been no reports of ART domains on PARPs bound to NAD^+^. However, in the recent structural characterization of PARP4 (vault PARP), a complex with NADH was revealed [78]. In this structure, the open conformation of HD makes it easy to access NAD^+^. PARPs have been known to modify proteins on not only on acidic residues but also on cysteine and lysine residues. 

Recently, an accessory factor, HPF1, was found to switch the amino acid specificity from aspartate or glutamate to serine residues [79,80]. Suskiewics et al. reported the co-structure of HPF1 bound to the catalytic domain of PARP2(CATΔHD) [81]. Subsequently, Sun et al. reported the structure of HPF1–PARP1(CATΔHD) [82]. These structural characterizations reveal that HPF1 forms a joint active site with PARP1 or PARP2. It was suggested that Glu284 in HPF acts as a key catalyst for Ser ADP-ribosylation. On the contrary, Glu988 in PARP is not a key residue for Ser ADP-ribosylation. Langelier et al. showed the structure of DNA-bound PARP1(Zn1, Zn2, WGR, CAT) and the DNA activation mechanism [83]. Furthermore, the cryo-EM structure of PARP2–HPF1–nucleosome showed three dynamic structures of the catalytic domain of PARP2 [84]. Little is known about how PARPs select specific amino acids to modify, though PARPs have been known to modify a variety of amino acids, such as Glu, Asp, Lys, and Ser [85]. Furthermore, the targeting of amino acids by mono-ADP-ribosylation PARPs (PARP6, PARP8, PARP10, PARP11, and PARP12) occurs through self-ADP-ribosylation on Glu, Asp, Lys, and Cys [86]. 

It is interesting to understand how these variabilities in targets are brought about. In summary, the specificities of PARPs are related to the configurations of their domains, their regulation, and their ability to recognize multiple substrates. Future work is expected to reveal the structures of PARP–substrate complexes.

## 8. Concluding Remarks

In ARTC, the spatial tripartite arrangement of the enzyme, NAD^+^, and substrate are conserved between protein-targeting ARTs and DNA-targeting ARTs. This suggests their common importance for ADP-ribosylation. In ARTD, the spatial arrangement of tripartite is similarly conserved with ARTC. However, further accumulation of experimental tripartite complex information is needed in both classes. Also, recent developments in accurate structure modeling, such as alphafold3, are poised to revolutionize our ability to predict the joint structures of complexes, including proteins, nucleic acids, and small molecules [87]. Combining experimental and AI-predicted structure-based approaches provides more information on tripartite structures, which is essential for further understanding the general ADP-ribosylation mechanism. 

ART inhibitor development is also essential in understanding the mechanism. However, in the case of ART, all known inhibitors mimic NAD^+^ as a cofactor. Examples include βTAD [88] or benzamide adenine dinucleotide [89], which inhibit all classes of ARTs, including bacterial ARTCs, ARTDs, and mammalian PARPs. In the case of PARPs, PARP inhibitors, such as Olaparib, are available. PARP1/2 inhibitors could be used as drugs in breast cancer treatment through a mechanism known as synthetic lethality [90]. Even in the case of PARPs, all clinically relevant PARP inhibitors are designed to mimic NAD^+^. As far as we know, no inhibitors fill the active space interacting with the substrate. In this context, tripartite complex structures are essential for future drug design. In summary, based on tripartite complex structures, further studies, such as QM/MM, can lead to the development of novel ART inhibitors. These integrative studies contribute to our understanding of the general ADP-ribosylation mechanism.

## Figures and Tables

**Figure 1 toxins-16-00313-f001:**
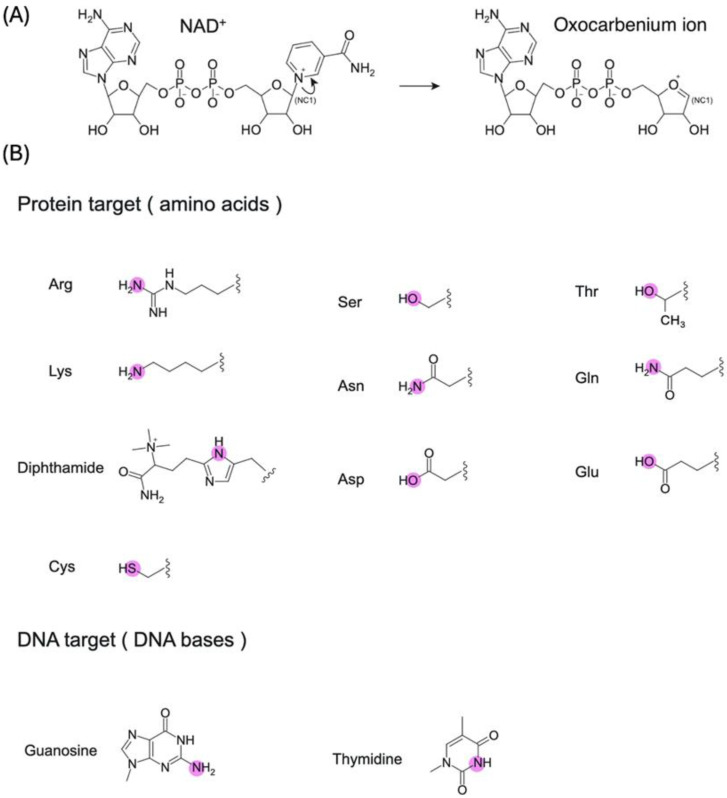
(**A**) ADP-ribosylation proceeds via an oxocarbenium ion after the cleavage of nicotinamide in NAD^+^. (**B**) ADP-ribosylation targets, specifically amino acids and DNA bases. The purple circles show the atoms that would be modified.

**Figure 2 toxins-16-00313-f002:**
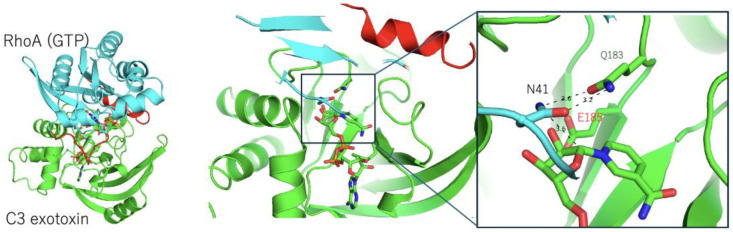
(**Left**) Structure of the C3 exotoxin (green) and the RhoA (GTP) (cyan) complex (PDB: 4XSH). The switch I and switch II regions are shown in red. (**Right**) A close-up view of the active site, showing 183-GlnXxxGlu-185 in C3 and Asn41 in RhoA. The key glutamate (Glu185) is labeled in red.

**Figure 3 toxins-16-00313-f003:**
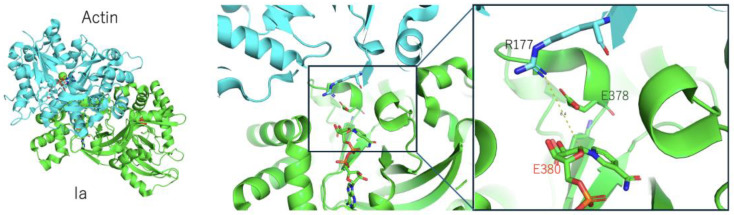
(**Left**) Structure of Ia (green) and the actin (cyan) complex (PDB: 4H03). (**Right**) A close-up view of the active site, showing 378-GluXxxGlu-380 in Ia and Arg177 in actin. The key glutamate (Glu380) is labeled in red.

**Figure 4 toxins-16-00313-f004:**
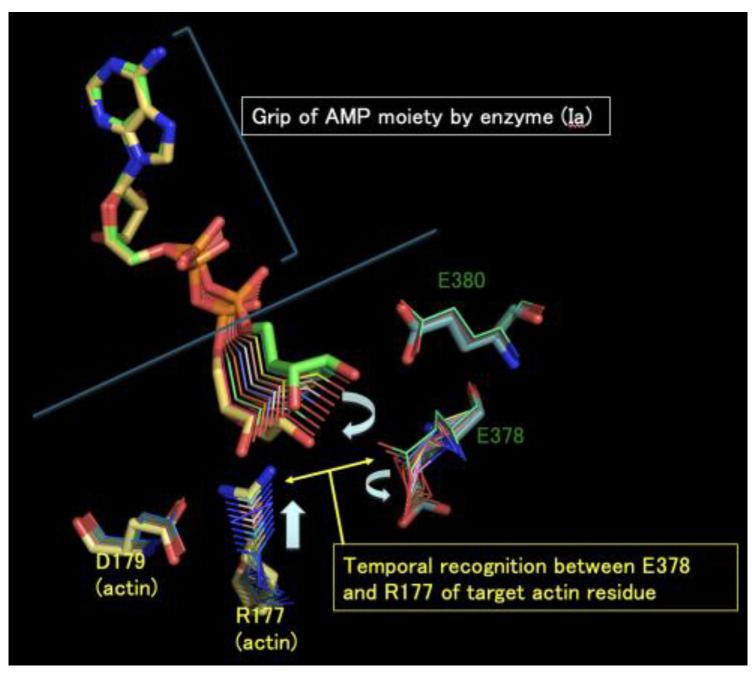
The strain-alleviation model of ADP-ribosylation. Structural changes in the active site and the oxocarbenium ion after cleavage of NAD+ are depicted. Green: immediately after the cleavage of NAD+. Yellow: just before the nucleophilic attack of Arg177. These models were modelled based on the crystal structures of pre- and post-ADP-ribosylation [46]. The detailed reaction is described in Chapter 2. Glu378 and Glu380 in Ia and Arg 177 and Asp179 in actin are labelled.

**Figure 5 toxins-16-00313-f005:**
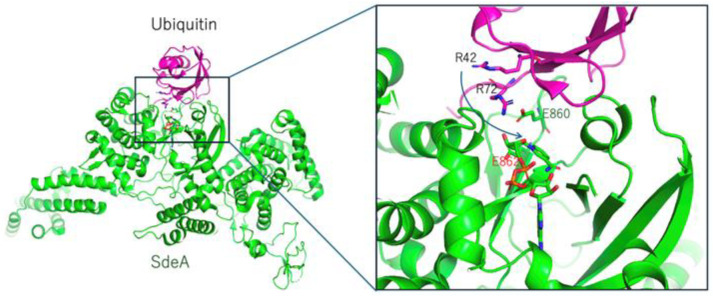
Structure of the SdeA (green) and ubiquitin (magenta) complex (PDB: 5YIJ). A close-up view of the active site, showing 860-Glu-Xxx-Glu-862 in SdeA and Arg42 (target) and Arg72 in ubiquitin. The key glutamate is labeled in red.

**Figure 6 toxins-16-00313-f006:**
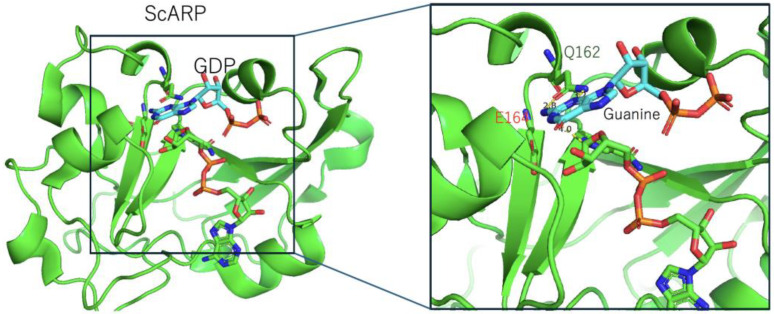
Structure of the ScARP (green) and GDP (cyan) complex (PDB: 5ZJ5). (Right) A close-up view of the active site, showing 162-Gln-Xxx-Glu-164 in ScARP and guanine in GDP. The key glutamate (Glu164) is labeled in red.

**Figure 7 toxins-16-00313-f007:**
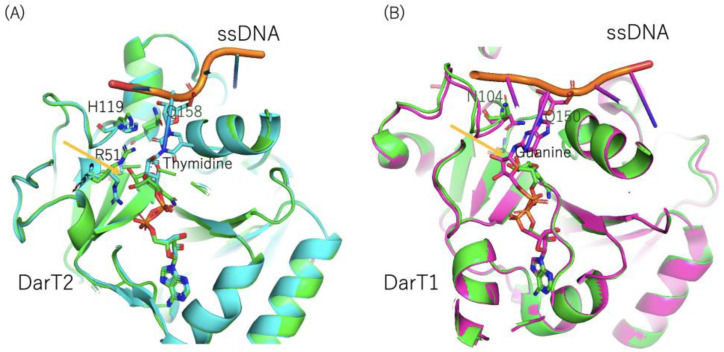
(**A**) Structures of DarT(T2)–ssDNA complexes (PDB:7OMW and 7ON0). The pre-reaction (green) and post-reaction (ADP-ribosylated) states (cyan) are shown, as are 158-Gln-Xxx-Glu(Glu160Ala)-160 in DarT2, His119, Arg51, and the target thymidine. The key glutamate (Ala160(Glu160Ala)) position is shown using an orange arrow. (**B**) Structures of DarT1–ssDNA complexes (PDB: 8BAR and 8BAQ). The pre-reaction (green) and post-reaction (ADP-ribosylated) states (purple) are shown, as are 150-Gln-Xxx-Glu(Glu152Ala)-152 in DarT1, Asn104, and the target guanine. The key glutamate (Ala152(Glu152Ala)) position is shown using an orange arrow.

**Table 1 toxins-16-00313-t001:** A list of complex structures of ADP-ribosyltransferase and its substrates that have been revealed using crystallography, including Ia-actin [25], C3–RhoA [26], ScARP-GDP [27], SdeA-ubiquitin [28,29], ExoA-eEF2 [30], DarT2-ssDNA [31], DarT1-ssDNA [32], and CteC-ubiqutin [33,34].

ART Class	Target	Residue or Base
**ARTC**		
Ia (Iota toxin A subunit)	actin	Arg 177
C3 exotoxin	RhoA	Asn 41
ScARP	GDP	guanine
SdeA	Ubiqutin	Arg 42
**ARTD**		
ExoA	eEF2	diphthamide 699
DarT2	DNA	thymidine
DarT1	DNA	guanine
**Chimeric featured ART**		
CteC	Ubiqutin	Thr 66

**Table 2 toxins-16-00313-t002:** ADP-ribosylating turn–turn loops in ARTC class enzymes. SdeA and pierisin family enzymes are shown in brown or cyan, respectively.

	Turn 1				Turn 2			
	X	X	*ϕ*	X	X	E/Q	X	E
Ia	P	G	Y	A	G	E	Y	E
C3cer	T	A	Y	P	G	Q	Y	E
SdeA	H	G	E	G	T	E	S	E
SdeB	H	M	T	G	S	E	D	E
SdeC	H	M	A	G	S	E	D	E
SidE	H	V	S	G	S	E	S	E
ScARP	H	K	W	A	D	Q	V	E
Scabin	H	K	W	A	D	Q	V	E
Pierisin-1	S	P	W	P	N	Q	M	E

## Data Availability

Not applicable.

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
