# Peer review of "General ADP-Ribosylation Mechanism Based on the Structure of ADP-Ribosyltransferase–Substrate Complexes"

_toxins, 2024, doi:10.3390/toxins16070313_

Round 1
Reviewer 1 Report
Comments and Suggestions for Authors
The work reviewed the reported mechanism for ADP-ribosylation through the substrate complex structure of ADP-ribosyltransferases. The manuscript should be significantly improved before consideration for publication.
1. The abstract and conclusion should be carefully improved. They are too short.
2. The previous works about the mechanism should be briefly discussed in Introduction.
3. The types of ADP-ribosyltransferases should be introduced and the mechanism should be discussed according to the difference types of ADP-ribosyltransferases.
4. How about the interactions between inhibitors and ADP-ribosyltransferases? This should be valuable for understanding the mechanism.
5. The techniques and methods used to investigate the mechanism should be discussed.
6. Figure captions in the figures should be removed.
7. The reference format should be carefully checked and revised.
8. Professional English presentation is required.
Comments on the Quality of English LanguageProfessional English presentation is required.
Author Response
Reviewer1
- The abstract and conclusion should be carefully improved. They are too short.
Thank you for your comments. We revised the summary as follows.
Abstract: ADP-ribosylation is a ubiquitous modification of proteins and other targets, such as nucleic acids, that regulates various cellular functions in all kingdoms of life. Furthermore, these ADP-ribosyltransferases (ART) modify a variety of substrates and atoms. It has been almost 60 years since ADP-ribosylation was discovered. Various ART structures have been revealed with the cofactor (NAD+ or NAD+ analog). However, we still do not know the molecular mechanisms of ART. It needs to be better understood how ART specifies the target amino acids or bases. For this purpose, more information is needed about the tripartite complex structures of ART, cofactor, and substrate. The tripartite complex is essential to understand the mechanism of ADP-ribosyltransferase. This review updates the general ADP-ribosylatioin mechanism based on the ART tripartite complex structures.
The concluding Remarks also was revised as follows.
8. Concluding Remarks
In ARTC, the spatial tripartite arrangement of the enzyme, NAD+, and substrate are conserved between protein-targeting ARTs and DNA-targeting ARTs. It suggests their common importance for ADP-ribosylation. In even ARTD, the spatial arrangement of tripartite is similarly conserved with ARTC. However, this experimental tripartite complex information needs to be further accumulated in both classes. Also, recently, the development in accurate structure modeling, such as alphafold3, will bring us a revolution capable of predicting the joint structure of complexes, including proteins, nucleic acids, and small molecules[87]. Using two approaches, experimental and AI-predicted structure-based, more information on tripartite structures is available. These are essential for further understanding of the general ADP ribosylation mechanism.
ART inhibitor development is also essential in understanding the mechanism. However, in the ART case, all known inhibitors are mimics of NAD+ as a cofactor. bTAD[88] or benzamide adenine dinucleotide[89] are inhibitors to inhibit all classes ARTs including bacterial ARTCs, ARTDs and mammalian PARPs. In PARP cases, PARP inhibitors such as Olaparib are available because PARP1/2 inhibitors could be a drug for the treatment of breast cancer through a mechanism known as synthetic lethality[90]. Even in the PARP case, all clinically relevant PARP inhibitors are designed to mimic NAD+. As far as we know, no inhibitors fill the active space interacting with the substrate. In this meaning, tripartite complex structures are essential for future drug design. In summary, based on tripartite complex structures, further studies such as QM/MM lead to the development of novel ART inhibitors. These integrative studies lead to the understanding of the general ADP-ribosylation mechanism.
- The previous works about the mechanism should be briefly discussed in Introduction.
I added next sentence in the introduction.
Several lines of evidence have shown that the ART reaction proceeds through an SN1-like transition state via an oxocarbenium ion (Figure 1A). In DT case, the transition state model most consistent with the isotope kinetic effects is characterized by the loss in bond order of the nicotinamide leaving group, suggesting a dissociative SN1 type reaction[16]. The oxocarbenium ion-like transition state is stabilized by Glu148, which is a key catalytic glutamate of DT[17]. The oxocarbenium ion-like transition state model is supported in the study of pertussis toxin[18].
- The types of ADP-ribosyltransferases should be introduced and the mechanism should be discussed according to the difference types of ADP-ribosyltransferases.
In both ARTC and ARTD, we consider the reaction via the oxocarbenium ion-like transition state. See also comment 2.
- How about the interactions between inhibitors and ADP-ribosyltransferases? This should be valuable for understanding the mechanism.
This is very good question. I added next sentence in concluding remarks.
ART inhibitor development is also essential in understanding the mechanism. However, in the ART case, all known inhibitors are mimics of NAD+ as a cofactor. bTAD[88] or benzamide adenine dinucleotide[89] are inhibitors to inhibit all classes ARTs including bacterial ARTCs, ARTDs and mammalian PARPs. In PARP cases, PARP inhibitors such as Olaparib are available because PARP1/2 inhibitors could be a drug for the treatment of breast cancer through a mechanism known as synthetic lethality[90]. Even in the PARP case, all clinically relevant PARP inhibitors are designed to mimic NAD+. As far as we know, no inhibitors fill the active space interacting with the substrate. In this meaning, tripartite complex structures are essential for future drug design. In summary, based on tripartite complex structures, further studies such as QM/MM lead to the development of novel ART inhibitors. These integrative studies lead to the understanding of the general ADP-ribosylation mechanism.
- The techniques and methods used to investigate the mechanism should be discussed.
As described in the comment 2, I revised and added the isotope kinetic effects techniques in the introduction. Please see comment 2.
- Figure captions in the figures should be removed.
OK
- The reference format should be carefully checked and revised.
OK
- Professional English presentation is required.
We did professional English editing at first and revised more.

Reviewer 2 Report
Comments and Suggestions for Authors
This is an interesting review on structural aspects of the ADP-ribosylation mechanism and substrate recognition of ADP-ribosyltransferases. While the topic and the information provided are relevant, I found it difficult to navigate through this review due to the way how the text is presented and to many language issues, more specifically grammar mistakes and the wording used in some sentences.
Here are some comments and suggestions:
-The abstract is very short and not very informative, it doesn’t do any justice to the contents of the review. Saying that we don’t know the mechanism and it needs to be better understood is not really informative. Moreover, this review is not about introducing structural approaches, it’s not a research paper. The abstract should be re-worded to better reflect the purpose of the review, and provide detail. About the stand of the field.
-In the introduction, the second sentence (Pathogenic bacteria…) should actually go first.
-In general, separate the text into paragraphs more often. Each section is a continuous chunk of text, which makes it quite difficult to navigate.
-Revise carefully the usage of acronyms. Very often they are not introduced in the right place (e.g DT, should go in line 31where it is mentioned first) or they are not used consistently, e.g ubiquitin in section 4 is introduced as Ub but then not used, then introduced again. Be also consistent with the usage of 3-letter names of aminoacids, e.g. in line 194 use Glu/Gln; there may be other places.
-There are plenty of instances where the wrong verb tense or form is used, I cannot list all of them, but for example in Fig. 1 legend, it should be Purple circles show the atoms that would be modifies, or to be modifies. In Fig. 2, 3, 5 it should be …are shown. (not …were shown.)
-Examples of sentences that need revision: line 49; line 87: what do you mean with later we focussed on it?; also a reference is missing. Lines 160-161 (incomplete sentence), 208 (you mean was identified as an ART…?) 259-260, 281 (“this was memorized as FAQ”; what does the author mean?), 287-288; 317-318; 319 (cause? brought about? Rather than brought)
-In general avoid using the first person throughout. For example, line 54, instead of In this review we…, change to The purpose of this review is…
Most notably in line 280, where the reference is to a paper with two authors. I suggest: In a previous review about… the general question about… was formulated. In general these few sentences at the beginning of section 7 should be re-written to provide a better flow.
-Enlarge Figure 1. Any chance you can show the oxocarbenium ion on panel A next to NAD+?
-In table 1 include a column with the references to publications on those structures. Heading: and its substrate revealed by crystallography. Why is PARP not included in the table?
-Line 111, introduce the bullet points (i), (ii) and (iii), they appear out of nowhere. How do they connect to the previous statement in lines 109-111?
-Line 120: Repalce “data no shown” by “our unpublished data”
-Fig. 3 legend, add: Key glutamate (E380) is labelled…
-Fig. 4. The legends needs more information about what is shown there.
-Lines 143, 145, who are “they”? Replace by It was suggested, it was proposed; or
-Line 172, you probably mean Table 2, not table 1.
-Lines 178-182, it’s not necessary to introduced quoted text, surely you can present that information using your own words.
-The concluding remarks are very cryptic and not very helpful. Not clear what the intellectual and physical divide between bacterial ARTs and PARPs is and what “from points to line” means. In this section I would expect a more informative wrap up about what has been achieved in recent years and a clearer outline about where the field is heading.
Other suggested changes:
Line 46, modify a variety
Line 73, reported the VIP2 structure
Line 108, for a direct attack
Line 121, remove distance
Line 147, replace watch by study or investigate?
Line 201 and 237, ADP-ribosylation of DNA…
Line 204, named pierisin
Line 211, pierisin1 to 6
Line 213, which are part of the ARTC class (you introduce ARTC and ARTD as classes, not families or subfamilies, same in line 244)
Line 244, and lacks activity on RNA
Line 246, provide the reference number for the Schuller study at the end of the sentence.
Line 247, italicize Thermus
Lines 278-279, has been summarized previously
Line 305, accessory
Line 308, you mean reported the HPF1… structure?
Line 311, on the contrary, Glu988 in PARP is not deprotonated
Line 315, known about how…
Comments on the Quality of English Language
The English language needs a thorough revision. At this stage the article is readable but below publication standard due to grammar mistakes, ill constructed sentences, incomplete sentences and wrong usage of some expressions. I recommend a thorough revision by a qualified English speaker.
Author Response
Reviewer 2
This is an interesting review on structural aspects of the ADP-ribosylation mechanism and substrate recognition of ADP-ribosyltransferases. While the topic and the information provided are relevant, I found it difficult to navigate through this review due to the way how the text is presented and to many language issues, more specifically grammar mistakes and the wording used in some sentences.
Here are some comments and suggestions:
We thank the reviewer for many basic points to improve our manuscript.
In this opportunity, we believe the manuscript was improved largely.
Also we used professional English correction service.
-The abstract is very short and not very informative, it doesn’t do any justice to the contents of the review. Saying that we don’t know the mechanism and it needs to be better understood is not really informative. Moreover, this review is not about introducing structural approaches, it’s not a research paper. The abstract should be re-worded to better reflect the purpose of the review, and provide detail. About the stand of the field.
We thank the reviewer for the point. We revised as follows.
Abstract: ADP-ribosylation is a ubiquitous modification of proteins and other targets, such as nucleic acids, that regulates various cellular functions in all kingdoms of life. Furthermore, these ADP-ribosyltransferases (ART) modify a variety of substrates and atoms. It has been almost 60 years since ADP-ribosylation was discovered. Various ART structures have been revealed with the cofactor (NAD+ or NAD+ analog). However, we still do not know the molecular mechanisms of ART. It needs to be better understood how ART specifies the target amino acids or bases. For this purpose, more information is needed about the tripartite complex structures of ART, cofactor, and substrate. The tripartite complex is essential to understand the mechanism of ADP-ribosyltransferase. This review updates the general ADP-ribosylatioin mechanism based on the ART tripartite complex structures.
-In the introduction, the second sentence (Pathogenic bacteria…) should actually go first.
Thank you for the comment. I exchanged.
-In general, separate the text into paragraphs more often. Each section is a continuous chunk of text, which makes it quite difficult to navigate.
Thank you for your comment. I separated text more throughout the manuscript.
-Revise carefully the usage of acronyms. Very often they are not introduced in the right place (e.g DT, should go in line 31where it is mentioned first) or they are not used consistently, e.g ubiquitin in section 4 is introduced as Ub but then not used, then introduced again. Be also consistent with the usage of 3-letter names of aminoacids, e.g. in line 194 use Glu/Gln; there may be other places.
> Thank you for your comments.
Especially, I checked ubiquitin (not Ub ) and DT (not diphtheria toxin) throughout the manuscript.
Amino acids: I changed to 3-letter names of amino acids.
-There are plenty of instances where the wrong verb tense or form is used, I cannot list all of them, but for example in Fig. 1 legend, it should be Purple circles show the atoms that would be modifies, or to be modifies. In Fig. 2, 3, 5 it should be …are shown. (not …were shown.)
Sorry for these corrections. In Fig. 1 legend, I reviseed “Purple circles show the atoms that would be modifies.”.
After English editing, I checked whole manuscript.
In Fig. 2, 3, 5, I revised to “are shown”.
-Examples of sentences that need revision: line 49;
>
How do ARTs select a wide variety of targets while also showing specificity toward many different atoms?
line 87: what do you mean with later we focussed on it?; also a reference is missing.
>
The sentence was revised as follows.
Our research group investigated this knowledge gap and revealed the structures of complexes with full protein substrates (C3-RhoA and Ia-actin) (Figure 2 and Figure 3)[25,26].
-Examples of sentences that need revision: line 49;
>
How do ARTs select a wide variety of targets while also showing specificity toward many different atoms?
line 87: what do you mean with later we focussed on it?; also a reference is missing.
>
The sentence was revised as follows.
Our research group investigated this knowledge gap and revealed the structures of complexes with full protein substrates (C3-RhoA and Ia-actin) (Figure 2 and Figure 3)[25,26].
Lines 160-161 (incomplete sentence)
>
However, bacterial pathogens have evolved elaborate mechanisms to hijack host Ub signaling pathways and evade the immune response. For this purpose, numerous E3 Ub ligases and deubiquitinating enzymes (DUBs) are encoded by bacterial and viral human pathogens.
208 (you mean was identified as an ART…?)
On the other hand, SCO5461 protein (ScARP) from Streptomyces coelicolor was identified as an ART that mainly targets mononucleotides and nucleosides and shares 30% homology with pierisin.
259-260,
I revised as follows.
Notably, this is the first study to report that ARTD also ADP-ribosylates guanine in a manner similar to ARTC enzymes, ScARP and pierisin.
281 (“this was memorized as FAQ”; what does the author mean?),
This question is answered as follows (Structural biology of ADP-ribosyltransferase-FAQ).
287-288;
317-318; 319 (cause? brought about? Rather than brought)
It is interesting to understand how these variabilities of targets are brought about.
-In general avoid using the first person throughout. For example, line 54, instead of In this review we…, change to The purpose of this review is…
Most notably in line 280, where the reference is to a paper with two authors. I suggest: In a previous review about… the general question about… was formulated. In general these few sentences at the beginning of section 7 should be re-written to provide a better flow.
>
I acknowledge the reviewer for the comment.
The purpose of this review is to compile relevant research related to ART–substrate complexes to gain insights into ADP-ribosylation substrate recognition mechanisms.
Several structures of PARPs have been revealed, but no structures have been available in the PARP complex with NAD+ (NADH) for a long time.
-Enlarge Figure 1. Any chance you can show the oxocarbenium ion on panel A next to NAD+?
Thank you. I revised the figure.
-In ine202 1 include a column with the references to publications on those structures. Heading: and its substrate revealed by crystallography. Why is PARP not included in the table?
I included the references. As far as we know, there is no tripartate complex structure of PARP at high resolution.
-Line 111, introduce the bullet points (i), (ii) and (iii), they appear out of nowhere. How do they connect to the previous statement in lines 109-111?
Thank you. OK, I changed.
-Line 120: Repalce “data no shown” by “our unpublished data”
I revised.
-Fig. 3 legend, add: Key glutamate (E380) is labelled…
I added this.
-Fig. 4. The legends needs more information about what is shown there.
I added as figure legend.
Figure 4. Strain-alleviation model of ADP-ribosylation. Structural changes of active site and oxocarbenium ion after cleavage of NAD+ are depicted. Green: immediately after cleavage of NAD+. Yellow: just before nucleophilic attack of Arg177. These models were modelled based on the crystal structures of pre- and post-ADP-ribosylation [46]. Detailed reaction was described in chapter 2. Glu378 and Glu380 in Ia, and Arg 177 and Asp179 in actin are labelled.
-Lines 143, 145, who are “they”? Replace by It was suggested, it was proposed; or
I revised as “The authors of this study”.
-Line 172, you probably mean Table 2, not table 1.
Thank you. I revised. new L202
-Lines 178-182, it’s not necessary to introduced quoted text, surely you can present that information using your own words.
I revised as follows “Of note, mutating Arg72 also completely abolished ubiquitin modification by SdeA. Accordingly, Arg42 and Arg72 mutants failed to be substrates of SdeA auto-ubiquitination. The author speculates that Arg72 plays a more important role in the catalysis by SdeA”.
-The concluding remarks are very cryptic and not very helpful. Not clear what the intellectual and physical divide between bacterial ARTs and PARPs is and what “from points to line” means. In this section I would expect a more informative wrap up about what has been achieved in recent years and a clearer outline about where the field is heading.
I revised the concluding remarks as follows.
8. Concluding Remarks
In ARTC, the spatial tripartite arrangement of the enzyme, NAD+, and substrate are conserved between protein-targeting ARTs and DNA-targeting ARTs. It suggests their common importance for ADP-ribosylation. In even ARTD, the spatial arrangement of tripartite is similarly conserved with ARTC. However, this experimental tripartite complex information needs to be further accumulated in both classes. Also, recently, the development in accurate structure modeling, such as alphafold3, will bring us a revolution capable of predicting the joint structure of complexes, including proteins, nucleic acids, and small molecules[87]. Using two approaches, experimental and AI-predicted structure-based, more information on tripartite structures is available. These are essential for further understanding of the general ADP ribosylation mechanism.
ART inhibitor development is also essential in understanding the mechanism. However, in the ART case, all known inhibitors are mimics of NAD+ as a cofactor. bTAD[88] or benzamide adenine dinucleotide[89] are inhibitors to inhibit all classes ARTs including bacterial ARTCs, ARTDs and mammalian PARPs. In PARP cases, PARP inhibitors such as Olaparib are available because PARP1/2 inhibitors could be a drug for the treatment of breast cancer through a mechanism known as synthetic lethality[90]. Even in the PARP case, all clinically relevant PARP inhibitors are designed to mimic NAD+. As far as we know, no inhibitors fill the active space interacting with the substrate. In this meaning, tripartite complex structures are essential for future drug design. In summary, based on tripartite complex structures, further studies such as QM/MM lead to the development of novel ART inhibitors. These integrative studies lead to the understanding of the general ADP-ribosylation mechanism.
Other suggested changes:
Line 46, modify a variety
Thank you. I changed as follows “target a variety of amino acids, nucleobases”.
Line 73, reported the VIP2 structure
Han et al. reported the structures of VIP2
Line 108, for a direct attack
suggesting a possible direct attack
Line 121, remove distance
I deleted the distance.
Line 147, replace watch by study or investigate?
it is still necessary to examine other ARTD complexes in the future.
Line 201 and 237, ADP-ribosylation of DNA…
OK
Line 204, named pierisin
OK
Line 211, pierisin1 to 6
OK
Line 213, which are part of the ARTC class (you introduce ARTC and ARTD as classes, not families or subfamilies, same in line 244)
OK ARTC class
Line 244, and lacks activity on RNA
and lacks activity on RNA or protein targets
Line 246, provide the reference number for the Schuller study at the end of the sentence.
The reference was added.
Line 247, italicize Thermus
OK
Lines 278-279, has been summarized previously
has been summarized in refs. [67,68].
Line 305, accessory
OK
Line 308, you mean reported the HPF1… structure?
Subsequently, Sun et al. reported the structure of HPF1–PARP1(CATΔHD)
Line 311, on the contrary, Glu988 in PARP is not deprotonated I changed as follows. It was suggested that Glu284 in HPF, which is the key for Ser ADP-ribosylation as a catalyst. On the contrary, Glu988 in PARP is not key residue for Ser ADP-ribosylation.
Line 315, known about how…
PARPs have been known to modify

Round 2
Reviewer 1 Report
Comments and Suggestions for Authors
Accept
Reviewer 2 Report
Comments and Suggestions for Authors
The authors have addressed all my critiques satisfactorily. The Engish language has also been improved considerably. I recommed publication of this review.